# Evaluation of an Ozone Chamber as a Routine Method to Decontaminate Firefighters’ PPE

**DOI:** 10.3390/ijerph182010587

**Published:** 2021-10-09

**Authors:** Marcella A. de Melo Lucena, Félix Zapata, Filipe Gabriel M. Mauricio, Fernando E. Ortega-Ojeda, M. Gloria Quintanilla-López, Ingrid Távora Weber, Gemma Montalvo

**Affiliations:** 1BSTR, Fundamental Chemistry Department, Federal University of Pernambuco-UFPE, Avenida Prof. Luiz Freire, S/N, CDU, Recife 50740-540, Brazil; 2Universidad de Alcalá, Departamento de Química Analítica, Química Física e Ingeniería Química, Grupo de Investigación CINQUIFOR, Ctra. Madrid-Barcelona km 33.6, 28871 Alcalá de Henares, Spain; felix.zapata@um.es (F.Z.); fernando.ortega@uah.es (F.E.O.-O.); 3Department of Analytical Chemistry, Faculty of Chemistry, University of Murcia, Campus Espinardo, 30100 Murcia, Spain; 4LIMA, Chemistry Institute, University of Brasilia-UNB, Brasilia 70904-970, Brazil; filipehades@gmail.com (F.G.M.M.); weber_it@hotmail.com (I.T.W.); 5Universidad de Alcalá, Departamento de Física y Matemáticas, Ctra. Madrid-Barcelona km 33.6, 28871 Alcalá de Henares, Spain; 6Universidad de Alcalá, Departamento de Química Orgánica y Química Inorgánica, Ctra. Madrid-Barcelona km 33.6, 28871 Alcalá de Henares, Spain; gloria.quintanilla@uah.es

**Keywords:** ozone, ozonolysis, firefighters, work safety, personal protective clothes (PPC), decontamination, polycyclic aromatic hydrocarbons (PAH)

## Abstract

Ozone chambers have emerged as an alternative method to decontaminate firefighters’ Personal Protective Equipment (PPE) from toxic fire residues. This work evaluated the efficiency of using an ozone chamber to clean firefighters’ PPE. This was achieved by studying the degradation of pyrene and 9-methylanthracene polycyclic aromatic hydrocarbons (PAHs). The following experiments were performed: (i) insufflating ozone into PAH solutions (homogeneous setup), and (ii) exposing pieces of PPE impregnated with the PAHs to an ozone atmosphere for up to one hour (heterogeneous setup). The ozonolysis products were assessed by Fourier Transform Infrared Spectroscopy (FTIR), Thin-Layer Chromatography (TLC), and Mass Spectrometry (MS) analysis. In the homogeneous experiments, compounds of a higher molecular weight were produced due to the incorporation of oxygen into the PAH structures. Some of these new compounds included 4-oxapyren-5-one (m/z 220) and phenanthrene-4,5-dicarboxaldehyde (m/z 234) from pyrene; or 9-anthracenecarboxaldehyde (m/z 207) and hydroxy-9,10-anthracenedione (m/z 225) from 9-methylanthracene. In the heterogeneous experiments, a lower oxidation was revealed, since no byproducts were detected using FTIR and TLC, but only using MS. However, in both experiments, significant amounts of the original PAHs were still present even after one hour of ozone treatment. Thus, although some partial chemical degradation was observed, the remaining PAH and the new oxygenated-PAH compounds (equally or more toxic than the initial molecules) alerted us of the risks to firefighters’ health when using an ozone chamber as a unique decontamination method. These results do not prove the ozone-advertised efficiency of the ozone chambers for decontaminating (degrading the toxic combustion residues into innocuous compounds) firefighters’ PPE.

## 1. Introduction

Among urban occupations, firefighting is by far one of the riskiest professions. Aside from potential injuries and fatalities [1], firefighters are exposed to a large variety of chemicals [2], including toxic fire residues that are associated with some types of cancer [3]. Several studies around the world evaluated the cancer incidence in firefighters, compared to their respective local population, indicating a significant excess of neoplasms, as lung, colon, kidney, among others, as well as cancer mortality itself [3,4,5,6,7,8,9].

It is known that fires generate toxic and carcinogenic substances as consequence of the incomplete combustion of wood, plastic, and other fuels present in structural or wildland fires [3,10,11,12,13,14,15]. Furthermore, the use of polymeric compounds in modern buildings and furniture can generate even more toxic substances [3,15]. Chemicals typically found in fires include, among others, aldehydes (mainly formaldehyde, and acrolein), aromatic compounds (as benzene), polycyclic aromatic hydrocarbons (PAH, as pyrene, anthracene, and benzo[a]pyrene), chlorinated hydrocarbons (as dichloromethane), as well as toxic inorganic gases (as CO, NO_x_, HCN, SO_2_, and HCl) known for causing acute effects [10,11,12,13,14,15,16,17,18,19,20,21].

Nowadays, firefighters rely on modern self-contained breathing apparatus which considerably minimizes the inhalation of toxic compounds while firefighting [16]. In addition, firefighters’ skin exposure to hazardous chemicals is partially protected by their uniforms, though they are still not fully impermeable to toxic organic chemicals as indicated by recent studies [22,23]. However, a secondary and important source of contamination after the fire is extinguished is the presence of combustion residues which remain impregnated on firefighters’ personal protective equipment (PPE) [24]. When they touch or wear these contaminated objects, toxic compounds can be absorbed through the skin or via inhalation [17,18]. In addition, these residues are normally transferred to fire trucks and fire stations, places where firefighters do not use PPE and are more unprotected and exposed. Therefore, unless properly decontaminated, the contaminated PPE accumulates toxic compounds, becoming a dangerous source of contamination [17].

The National Fire Protection Association (by the NFPA 1851 guideline) establishes three cleaning levels for the PPEs [25]. The routine cleaning (or field decontamination) should be performed after each intervention and should not result in the equipment becoming out of service. Usually, this cleaning includes brushing and/or rinsing the PPE with water. The advanced treatment is performed by the appropriate washer extractors and should be carried out at least twice a year. Thirdly, the most specialized treatment is used for the decontamination of specific chemical or biological hazardous products [15,25]. Although the advanced cleaning is more efficient for removing contaminants than the routine cleaning, it can damage the PPE and endanger the firefighters’ safety [15]. Furthermore, the advanced cleaning needs to remove the PPE from service for longer. Therefore, it is important to invest in methods to improve the efficiency of the routine cleaning.

Fent et al. [17], evaluated the efficiency of three routine cleaning procedures in removing PAH from the firefighters’ PPE: (i) an industrial scrub brush to remove debris (dry-brush decontamination); (ii) an air jet under the PPE (air-based decontamination); and (iii) spraying the PPE with a mixture of water and soap, followed by the industrial scrub brush decontamination, and water rinsing until the residues are no longer visible (wet-soap decontamination). The latter was the most efficient procedure, reducing 85% of the PAH, while the dry-brush and the air-based procedures reduced only 23% and 2%, respectively. Similar results regarding the incomplete removal of PAHs and other semi-volatile organic compounds through these decontamination and laundering methods were recently corroborated by Banks et al. [26]. However, although more efficient, the wet-soap decontamination is time consuming and would take the PPE out of service while drying, making it difficult to apply it after each intervention. This is the reason why, although firefighters are aware of the risk of developing cancer, recent research shows that less than 20% of them clean their PPE after each use, due to the lack of time between calls and the discomfort of wearing wet PPE [27,28]. In this sense, alternative and faster methods to improve the efficiency of the routine PPE cleaning are needed.

An emergent method to decontaminate the firefighters’ PPE is the use of the ozone chambers. Its efficiency is typically reported by the manufacturer as the elimination of toxic volatile fractions from the air resulting from the evaporation of volatile aromatic compounds, such as toluene. According to different manufacturers’ information, the volatile fraction is reduced in large percentages in just 2–5 min, as a consequence of its degradation by ozone and complete oxidation to CO_2_ and water. This way, in the ozone PPE decontamination approach, the PPE would be put inside a cabinet and exposed to an ozone atmosphere for a few minutes. After that, the PPE would be ready for use again. Therefore, this method would not require any effort from the firefighters and would be easily applied after each use, appearing to be an easy, fast, and promising routine decontamination method. However, despite these great advertised advantages, there are still no studies that prove its efficiency for cleaning the wide range of combustion products on firefighters’ PPE.

Ozone is a strong oxidizer used to treat waste and drinking water [29], in food disinfection [30], and as a disinfectant in hospitals [31]. In addition, the ozonolysis reaction is typically used in organic synthesis to break down organic molecules by means of the reaction between ozone and the double/triple bonds [32]. However, long reaction times, a high temperature, or the use of catalysts are usually required to achieve significant reaction yields. The simple approach (ozone exposure), proposed by the ozone chamber manufacturers to achieve such a high degradation yield of the aromatic compounds in such a short time, must be scientifically studied. In addition, it must be specifically tested for the decontamination of the low, volatile, toxic, organic fire residues, including PAH, on the firefighters’ PPE.

Therefore, the first objective of this work was to evaluate the PAHs degradation in the liquid phase (homogeneous set up) under a mild temperature and time conditions. Secondly, the efficiency of the ozone chamber was assessed by using the firefighters’ personal protective clothing (PPC) contaminated ad hoc with PAHs (heterogeneous set up). Pyrene and 9-methylanthracene were selected as PAHs in this study because both (pyrene and substituted-anthracene) were experimentally verified as toxic compounds produced in fires. This was revealed not only in the literature [12,13,14,19], but also in an on-going experimental study performed by the authors (in which firefighters exposed to different fires were analyzed to qualitatively determine the toxic substances on their PPE). To the best of the authors’ knowledge, this was the first time that the efficiency of an ozone chamber to degrade toxic PAHs in firefighters’ PPE was evaluated by an independent institution with no commercial interests.

## 2. Materials and Methods

Pyrene (98%, Alfa Aesar) and 9-methylanthracene (99%, Alfa Aesar, Haverhill, MA, USA) were used as purchased, without any purification process. Toluene (99.8%, Sigma Aldrich, St. Louis, MO, USA), dichloromethane (99.8%, Sigma Aldrich), and acetonitrile (99.5%, Merck, Kenilworth, NJ, USA) were used as solvents. The ozone exposure experiments were performed using a commercial Ozone Chamber from the Fire Department of the Comunidad de Madrid. To assess the degradation efficiency, two types of experiments were performed: homogeneous (PAHs in solution) and heterogeneous (pieces of PPC impregnated with PAHs) schemes.

### 2.1. Homogeneous Experiments

A two-neck, round-bottom flask worked as a reactor and received 40 mL of 0.2 M solution of pyrene or 9-methylantracene in acetonitrile. A high PAH concentration was used in these experiments to clearly verify whether the degradation facilitated by ozone was significant, and to detect the generated by-products more easily (i.e., a higher concentration of by-products will be obtained when starting from a high concentration of PAH). As shown in the Appendix A, flexible tubes were used to connect the first neck of the flask with the ozone inlet, whereas the second neck was connected to a cold trap system (immersed in liquid nitrogen) to avoid the loss of any volatile compound. After different exposure times (15, 30, 45, or 60 min) to the bubbling ozone, the sample (and the blanks) aliquots were collected and analyzed by infrared spectroscopy (IR), thin layer chromatography (TLC), and mass spectrometry (MS) to identify the possible degradation products. To minimize the dragging of the PAH caused by the acetonitrile evaporation, these experiments were subsequently replicated using toluene as solvent. All experiments were performed in triplicate.

### 2.2. Heterogeneous Experiments

This experiment was conducted using fragments of new synthetic fabrics identical to those used in the manufacture of the Madrid firefighters’ PPC (kindly supplied by Protec Solana SL). These fabrics were composed of three layers: the outer shell 220 g (99% aramid and 1% carbon fibre), the moisture barrier 140 g (PTFE-laminated aramid), and the internal thermal liner 130 g (40% aramid, 40% viscose, and 20% Zylon). The fabrics were cut into 2 × 2 cm fragments and sewn on the four edges to keep the layers together (Appendix A), in the same order as the official protective clothing of the firefighters.

A 0.01 M pyrene solution and a 0.05 M 9-methylanthracene solution were prepared in dichloromethane (99%, Scharlau). Then, 100 μL of each solution were slowly dripped on the outer shell of the 60 PPC fragments (30 fragments for each PAH compound). These fragments were left to dry for 24 h at room temperature, and then they were exposed to ozone in the chamber. The fabrics were kept in the ozone chamber for 0, 5, 15, 30 or 60 min (three fragments for each time interval). A similar set of experiments was performed using moistened conditions. In those experiments, 60 PPC PAHs-impregnated fragments were sprayed with distilled water immediately before the ozonation. All other procedures were identical to those performed on the dry PPC fragments. After the ozonation, the PPC fragments were first analyzed by FTIR. Then, they were placed into 10 mL plastic tube containing 10 mL of acetonitrile or dichloromethane for extraction. Both solvents were evaluated. Each tube was placed into an ultrasonic bath (Elma D-7700 Singen Transsonic Digital) operating at 35 kHz for 10 min at 40 °C. The liquid was then transferred to a new plastic tube and the fabric fragment was placed inside a plastic syringe and pressed by the plunger to maximize the liquid extraction. All samples were then analyzed by UV-Vis spectroscopy, and TLC. The initial and final samples (0 and 60 min) were also analyzed by mass spectrometry.

### 2.3. Instrumentation

A Lambda 35 spectrophotometer (PerkinElmer, Waltham, MA, USA) operating with double beam and equipped with deuterium and tungsten lamps was used to perform the UV-Vis measurements. All samples were diluted (pyrene samples: 350 μL of sample in 3 mL of solvent; 9-methylanthracene samples: 50 μL of sample in 3 mL of solvent) so that the maximum absorbance for the untreated samples (0 min samples) were close to 1. All spectra were collected in the 200 to 700 nm range.

The IR measurements were performed using an FTIR Nicolet TM iSTM10 TM spectrometer (Thermo Fisher Scientific, Waltham, MA, USA), equipped with an attenuated total reflectance (ATR) accessory. The FTIR spectra were collected in the 4000 to 650 cm^−1^ range, with a 4 cm^−1^ resolution, and underwent 64 scans, using the OMNIC 9.0 software (Thermo Fisher Scientific, Waltham, MA, USA). The IR spectra were acquired from the pure solid PAHs, the PAH-solutions (after the solvent evaporation), and before and after the ozonation for both, the homogeneous and heterogeneous experiments.

The TLC analysis were carried out on 60 Silica gel TLC plates coated with a fluorescent indicator F254 (Merck & Co, Kenilworth, NJ, USA) using hexane:diethylether (80:20, *v*/*v*) as eluting solvent. After the migration, the TLC plates were visualized using an UV lamp working at 254 nm.

The mass spectra were acquired using a 6210 Time-of-Flight Mass Spectrometer system (Agilent Technologies, Santa Clara, CA, USA) equipped with an Atmospheric Pressure Chemical Ionization (APCI) interface. The APCI procedure was performed in both positive and negative ion modes. The samples were directly injected (1 μL) into the mass spectrometer (the solid standards were solved and diluted in acetonitrile), without performing a previous chromatographic separation. The system was set to scan in the 100 to 1500 m/z range. The Agilent Qualitative Analysis B.07 software (Agilent Technologies, Santa Clara, CA, USA) was used for the data processing.

## 3. Results and Discussion

The ozone chamber manufacturers take for granted and/or advertise that a high degradation percentage of the aromatic compounds takes place in a few minutes under an ozone exposure, which might be supported by the decrease in the concentrations of these compounds. This decrease is attributed to the conversion of the original monocyclic aromatic compounds into CO_2_ and H_2_O. This assumption is questionable because previous studies show that the aromatic compounds suffer a slow and probably partial degradation by ozone, usually requiring long ozonation times, high ozone dosages, aqueous solutions, acid/alkaline media, and/or the use of catalysts [33,34,35,36]. Therefore, our hypothesis states that the chemical degradation is not exclusively responsible for the decrease in the aromatic compounds’ concentrations, but also the physical dragging of the ozone flow.

When turning on the ozone flow, the atmosphere inside the chamber becomes richer in ozone. Given the fact that the chamber is not hermetic, but has plenty of non-welded joints and small open edges, a positive air flow would be created from the inside to the outside of the chamber. Consequently, the ozone flows from the inside to the outside of the chamber displacing any other gases, which include the vapour phase of the aromatic compounds. This way, if we initially have an equilibrium between the liquid and vapour phases of the compounds, then the vapour concentration is modified when the ozone flow is activated because some of the vapour is rapidly expelled outside of the chamber. Thus, some molecules from the liquid compounds would have to be converted again into vapour to try to reach a new equilibrium. Hence, the amount of both the liquid and the vapour concentrations inside the chamber would become smaller and smaller with time. Therefore, the decrease in the aromatic compounds’ concentration would not be exclusively due to the chemical degradation, but mostly due to the dragging effect accelerated by the ozone flow from the inside to the outside of the chamber.

Accordingly, to verify the effectiveness of the ozone chemical degradation process and to discard other processes, two experiments were proposed in this study: (i) studying PAHs in solution (homogeneous experiment); and (ii) studying PAHs absorbed on new synthetic fabric fragments of the Madrid firefighters’ PPC (heterogeneous experiment).

### 3.1. Homogeneous Experiments

The visual most evident change among the first experiment set-up was the decrease in the volume of the PAH-acetonitrile solution contained inside the flask. After 60 min of ozone bubbling, the volume of the solution inside the flask was reduced to approximately one tenth of the initial volume. As expected, the rest of the solution was found frozen inside the cold trap (set at the end of the tubbing path). Thus, the use of a cold trap system allowed the observation of the great dragging effect of the intense ozone flow, which might explain the large decrease in the volume of the PAHs solution detected in the experiment. Nonetheless, the physical dragging was not incompatible with a simultaneous chemical degradation. Therefore, the PAHs chemical degradation that might potentially occur in the solution (besides the dragging) was evaluated by different analytical techniques.

After checking that the acetonitrile solvent was not chemically affected at all by the ozone, the PAHs solutions in the acetonitrile were studied. In this respect, the FTIR analysis revealed some modifications in the pyrene and 9-methylanthracene FTIR spectral bands, suggesting that they underwent some kind of chemical reaction. The products presented carbonyl groups as evidenced by the FTIR bands located at 1740 and 1652 cm^−1^ for the ozonized 9-methylanthracene, and at the 1683 cm^−1^ band for the ozonized pyrene (Figure 1). The other new bands that appeared in the 9-methylanthracene solution (but not in the pyrene solution) after the ozonation seemed to reveal that the 9-methylanthracene was far more degraded than the pyrene. However, in both cases, the degradation was not complete because the pure pyrene and 9-methylanthracene bands were still present.

In order to reduce the dragging effect, the experiment was repeated after replacing the acetonitrile by toluene, which had a lower volatility. This time, only about 10% of the solution volume was evaporated and dragged towards the cold trap. Unlike acetonitrile, toluene was an aromatic hydrocarbon that might be affected by ozone. However, and unexpectedly, no spectral difference was observed in the pure toluene FTIR spectrum before and after the ozonation (Appendix A). It was known from the literature that toluene could be partially degraded by ozone in such a way that the ozone could be used to remove toluene from water [37]. However, according to the literature, a significant toluene degradation requires the combination of water, ozone, and some kind of catalyst [37].

Regarding the PAHs, the Appendix A shows the FTIR spectra of the pyrene and 9-methylantracene-toluene solutions (before and after the ozonation). The overlapping toluene bands hindered the identification of the corresponding pyrene and 9-methylantracene spectral bands, and/or the potential new compounds produced by the ozonolysis. Nevertheless, it was still observed that the ozone degraded more 9-methylanthracene than pyrene. It should be remarked that apparently the pure toluene was not degraded by the ozone. Therefore, the new bands were associated with the partial oxidation of the PAHs.

In addition, various aliquots of the toluene solutions containing PAHs were analyzed by TLC after different ozonation times. The TLC plates revealed the appearance of some byproducts after the ozonation, though pyrene and 9-methylanthracene were still present in large amounts (Figure 2).

Mass spectrometry was used to identify these by-products. Figure 3 shows the mass spectra acquired in the homogeneous experiments. The molecular ion [M-H]^+^ of each compound was detected because the soft ionization method (APCI) does not normally cause too much fragmentation of the compounds [38]. The spectrum of the pure pyrene standard (Figure 3a) showed the presence of a single peak with m/z 203, which corresponded to the [M-H]^+^ ion. The ozonized pyrene spectrum (Figure 3b) also presented the m/z 203 signal, in addition to four new peaks (m/z 191, 205, 221, and 235). Unexpectedly, most of the peaks showed a higher m/z ratio, which suggests that the oxygen atoms were incorporated into the product’s structures, producing oxygenated PAH compounds. In fact, three of these compounds were previously identified in literature as 4-oxapyren-5-one (m/z 220) [39,40,41,42], phenanthrene-4,5-dicarboxaldehyde (m/z 234) [39,40,41,42], and cyclopenta[def]phenanthren-4-one (m/z 204) [43] (values of the [M-H]^+^ species). The fourth compound was not identified but, similar to the others, is most likely had a phenanthrenic structure (m/z > 178) [39].

In pyrene, the ozone attack starts preferentially through the 4,5- and 9,10- bonds (called the K region), since these bonds have a higher double-bond character and, as consequence, they are more reactive [41,43,44]. According to the literature, phenanthrene-4,5-dicarboxaldehyde (m/z 234) is one of the first pyrene degradation intermediates, while cyclopenta[def]phenanthren-4-one (m/z 204) [43], and 4-oxapyren-5-one (m/z 220) [39,41] are produced by the subsequent ozonolysis of the former. In this work, no lower m/z peaks were observed (considering that the lowest detection limit was m/z 100).

It is important to highlight that ozonation is a complex reaction that can produce several different products. For instance, Yao et al. [40] reported 14 different products from the pyrene ozonation in an acetonitrile/water (90/10) solution, all presenting aromaticity. In turn, Zeng et al. [43] mentioned that the pyrene ozonation in water produced a series of oxygenated and long-chain aliphatic hydrocarbons. These differences may be associated with the amount of ozone (which can lead to a subsequent breakdown of the intermediates) and the solvents used in the reaction. In water, the ozone self-decomposes into OH-radicals, which will participate in the oxidation process and promote a higher decomposition. The results of the present work (Figure 3) showed that, although the ozone was able to open the pyrene ring, the amount of ozone provided by the chamber in one hour was not enough to fully degrade the 8 mmol of pyrene present in the solution.

In the case of 9-metylanthracene, the untreated standard (Figure 3c) showed three intense peaks at m/z 178, 193, and 207. The first belonged to the anthracene [M]^+^, the second belonged to the pure 9-methylanthracene [M-H]^+^, and the third belonged to the 9-anthracenecarboxaldehyde [M-H]^+^. The anthracene and 9-anthracenecarboxaldehyde were known impurities that should not have interfered in the results. In turn, the ozonized sample (Figure 3d), as well as these three signals, showed a new and higher peak at m/z 225, which was attributed to the hydroxy-9,10-anthracenedione also reported in the literature [45]. Further, the intensity of the m/z 207 peak, attributed to an oxygenated-PAH, increased in relation to the 9-methylanthracene peak due to the oxidation of this compound. As in pyrene, the ozonation products were oxygenated polycyclic aromatic structures (containing aldehydes, ketones, or lactones), and, again, the initial compounds were not completely degraded after a one-hour treatment.

It is important to mention that when a PAH is absorbed by the organism, it is generally metabolized into PAH oxides. These oxides can form stable DNA-adducts (mainly with guanine and adenines) and induce mutations that are strongly associated with the process of cancer formation [3]. In addition, some studies suggest that the oxygenated-PAH produced by the partial PAHs oxidation may be equally or even more toxic than PAHs for both humans and the environment [46]. Thus, although the formation of species smaller than m/z 100 cannot be discarded (since the lowest limit measured was m/z 100), the oxygenated PAH detected in these experiments triggers an alert for the ozone-treatment potential risks to the firefighters and the environment.

### 3.2. Heterogeneous Experiments

The ozone-involving reactions are quite complex and sensitive to the reaction parameters mentioned above. Considering this, and for simulating a more realistic scenario, the heterogeneous experiments involved impregnating PAHs on the firefighters’ PPC, followed by the subsequent exposure to an ozone atmosphere, as recommended by the chamber manufacturer. Figure 4 shows the UV-Vis spectra of the products extracted from the heterogeneous experiments. The pyrene spectra (Figure 4a) show three intense bands with the maximum at 242, 272, and 336 nm wavelengths. These bands were attributed to the π-π* pyrene transitions (S0→S2, S0→S3, and S0→S4 transitions) [47]. The 9-methylanthracene UV-Vis spectra (Figure 4b) showed an intense band centered at 253 nm, as well as a set of bands associated with the S0→S1 transitions between 300 and 400 nm [48]. In both spectra, an increase in the ozone exposure time rendered a reduction in the intensity of the pyrene and 9-methylanthracene absorption bands. This indicated a decrease in the PAHs concentration (either because of the physical dragging effect and/or because of a chemical degradation). Additionally, the UV-Vis spectra also showed a greater reduction in the concentration of 9-methylanthracene than in pyrene, which was consistent with the FTIR results from the homogeneous experiments.

Furthermore, both compounds presented wavelength regions where the absorbance increased a little over time (tagged by blue rectangles in Figure 4). This fact might suggest that some byproducts were formed during the ozonolysis. However, the absorbance due to these new products was almost negligible in comparison to the absorbance observed for PAHs. In addition, a similar set of experiments were performed on PPC fragments with pyrene and 9-methylantracene using moistened conditions. This was intended to evaluate the influence of water, also simulating a wet PPC after a firefighting intervention. Similar results were obtained using either the dry or moistened conditions (Appendix A).

The FTIR analysis of the ozonized PPC fragments first evidenced that the PPC material (mostly composed of polyamides) was not chemically affected by the ozone (under the tested conditions and time interval). Unfortunately, the FTIR analysis did not provide further information about the PAHs chemical oxidation. This was most likely because the characteristic bands of the PPC (polyamides) outer layer overlapped with any of the characteristic bands of the 9-methylanthracene, pyrene or their potential by-products after the ozonation (Appendix A). The complex spectrum of the PPC outer layer hindered the detection of any weak band of either the PAHs or the potential ozonation byproducts. Similarly, no oxidation products were detected by the TLC analysis, but only the original PAHs (Appendix A). This result indicated that either no degradation occurred at all or that the partial oxidation was significantly lower than in the homogeneous experiment. This may have occurred in such a way that the little concentration of the ozonation products was insufficient to be detected by the macroscopic TLC.

Since FTIR and TLC techniques did not present high sensibility, the material extracted from the PPE fragments was also analyzed by mass spectrometry. Figure 5 shows the MS spectra of the samples treated at 0 and 60 min. All samples showed peaks attributed to some products extracted from the PPC, such as dyes and additives (m/z 177, 199, and 226). Appendix A shows the MS spectrum of the untreated PPC extract, which worked as blank reference. The pyrene-containing samples, after the ozone treatment, still showed the m/z 203 peak corresponding to pyrene. This would imply that the samples were not significantly degraded by the ozonolysis, as observed by UV-Vis, FTIR, and TLC. However, four new peaks were observed (m/z 191, 205, 221, and 235), which were the same previously identified in the homogeneous experiment. As mentioned, they corresponded to the pyrene oxidation products that had a phenanthrenic structure and were mostly heavier than the original compound.

In the samples with 9-methylanthracene, the m/z 178, 193, and 207 peaks corresponding to the 9-methylanthracene (and to the anthracene and 9-anthracenecarboxaldehyde impurities) were present in both the treated and untreated samples, also in accordance with the homogeneous experiment. However, after the ozone treatment, the relative intensities of the anthracene (m/z 178) and 9-methylanthracene (m/z 193) peaks decreased, while the relative intensity of the 9-anthracenecarboxaldehyde increased (m/z 207), and a new peak appeared (m/z 225), which would correspond to the hydroxy-9,10-anthracenedione. The experiments performed in moistened conditions generated the same products (Figure 5).

These results showed that, although the use of an ozone chamber was a relatively fast method which did not require effort for the firefighters, it was not efficient to clean the PPE because a significant amount of PAHs still remained even after one hour of treatment. Moreover, the products generated by the PAH’s partial ozone-induced oxidation could be equally or more toxic to humans and to the environment than the initial PAH compounds [46]. Aside from the fact that further experiments with longer treatment times are necessary, the formation of oxygenated PAH can be a dangerous and limiting factor for the use of ozone chambers as unique routine cleaning procedure.

## 4. Conclusions

This work studied the efficiency of the ozonolysis as a method to decontaminate PAH on firefighters’ personal protective clothes. All experiments were performed with pyrene and 9-methylanthracene, which were compounds that might be present in fire residues and hence, in firefighters’ contaminated PPE. In the homogeneous phase, the FTIR, TLC and MS techniques revealed the partial PAH degradation by ozone, whereas only UV-Vis and MS were able to show some partial degradation in the heterogeneous experiments. This result clearly demonstrates that a proper interaction between the ozone and the PAH reactant was required for the ozonolysis reaction to be significant, as occurs when bubbling up ozone into a PAH solution. However, the ozonolysis reaction was barely carried out when the reaction was conducted in an ozone atmosphere, directly placing the PPC fabrics contaminated with the PAH residues. Furthermore, both experiments (homogeneous and heterogeneous systems) showed that the new oxygenated PAH compounds were produced. These compounds could be similarly or more toxic to humans and more harmful to the environment compounds than the original PAH. In summary, this study did not demonstrate the ozone’s treatment efficiency for degrading the toxic combustion residues from PPE into innocuous compounds. Hence, this work suggests a further investigation into the use of ozone chambers intended to be used as a preventive measure to clean and decontaminate the firefighter’s PPE. Among the remaining limitations, the level of degradation using ozone should be determined for a higher number of organic residues from fires, including not only other PAHs, but also aldehydes, chlorinated hydrocarbons, and any other common toxic and carcinogenic substances from fires. In addition, an improved method that provides a better contact between ozone and contaminated PPE should be researched because the simple exposure of PPE to an ozone atmosphere was revealed to be insufficient.

## Figures and Tables

**Figure 1 ijerph-18-10587-f001:**
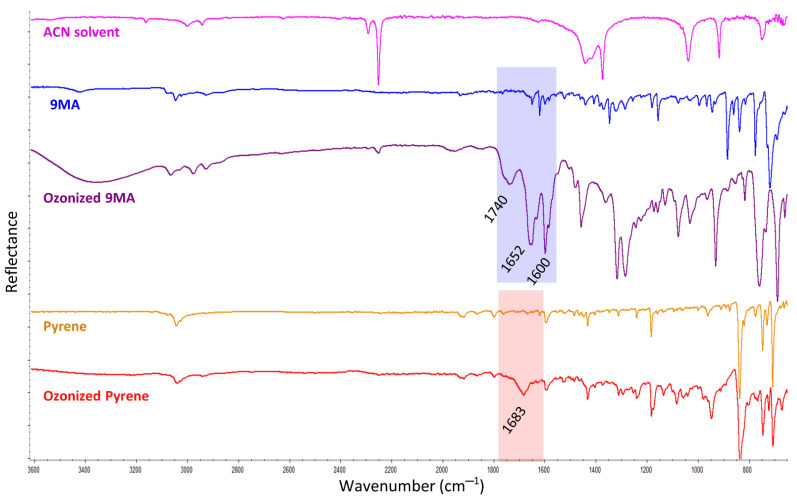
FTIR spectra of the 9-methylantracene (9MA) and pyrene standards in acetonitrile solutions (after the solvent evaporation on the ATR), and after 60 min of ozone treatment in the homogeneous experiments (also after the solvent evaporation on the ATR). The acetonitrile solvent (ACN) spectrum is also displayed as reference. The FTIR spectra are vertically offset for clarity.

**Figure 2 ijerph-18-10587-f002:**
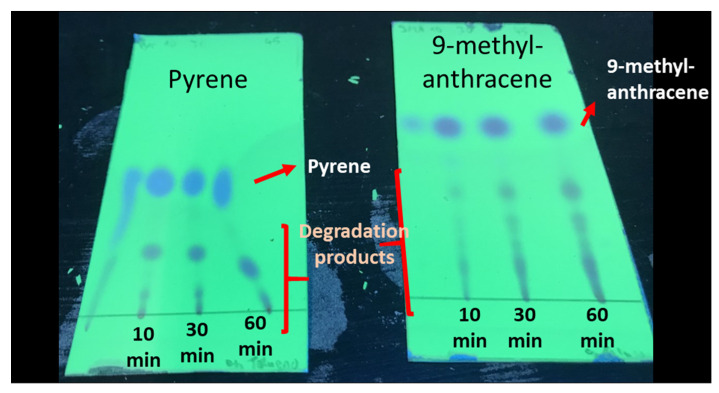
TLC chromatograms of the pyrene (left) and 9-methylantracene (right) toluene solutions treated by ozone in the homogeneous experiments for 10, 30 and 60 min. The images were acquired under UV radiation (254 nm).

**Figure 3 ijerph-18-10587-f003:**
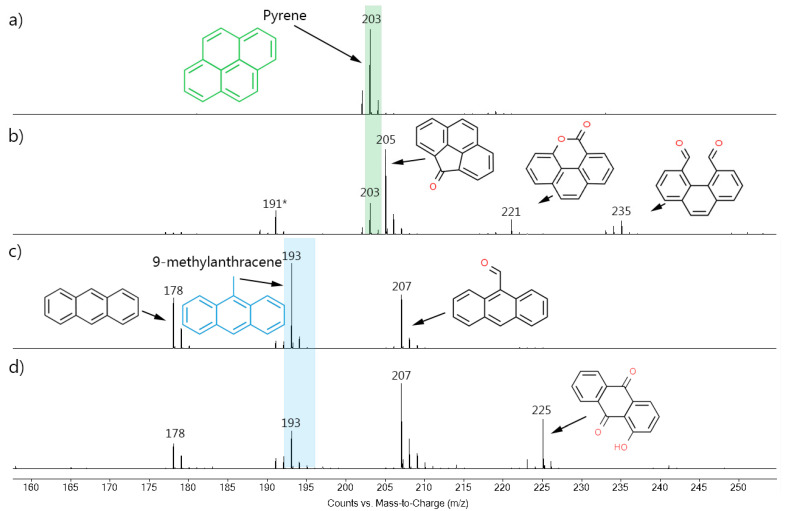
MS spectra of the (**a**) untreated and (**b**) treated pyrene; and of the (**c**) untreated and (**d**) treated 9-methylanthracene, both in the homogeneous ozonolysis experiments performed using acetonitrile as solvent. Green and blue rectangles indicate pyrene and 9-methylanthracene signals, respectively. * indicates unidentified structure.

**Figure 4 ijerph-18-10587-f004:**
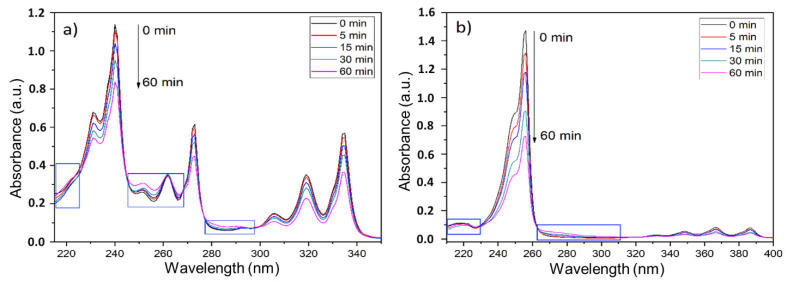
UV-Vis absorption spectra of the samples extracted from the PPC fragments with (**a**) pyrene and (**b**) 9-methylanthracene, after different ozonation times.

**Figure 5 ijerph-18-10587-f005:**
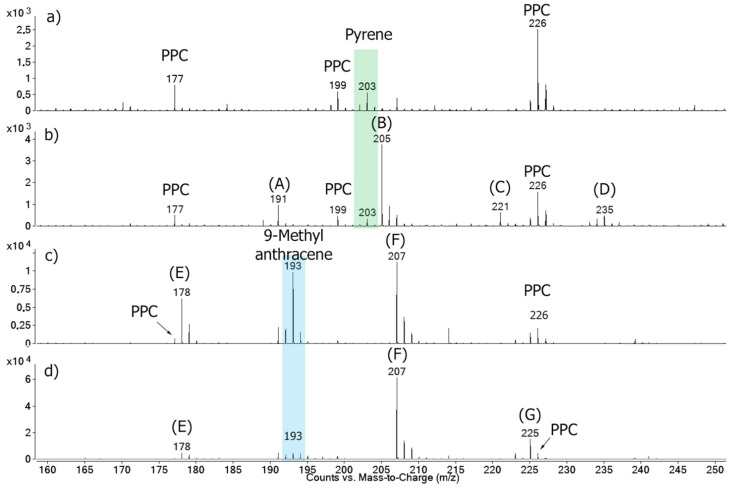
MS spectra of the (**a**) untreated and (**b**) treated PPC impregnated with pyrene; (**c**) untreated and (**d**) treated PPC impregnated with 9-methylanthracene; all after the heterogeneous experiments performed under dry conditions. In figure: (A) Unidentified phenanthrenic structure; (B) cyclopenta[def]phenanthren-4-one [43]; (C) 4-oxapyren-5-one [39,40,41,42]; (D) phenanthrene-4,5-dicarboxaldehyde [39,40,41,42]; (E) anthracene; (F) 9-anthracenecarboxaldehyde; and (G) hydroxy-9,10-anthracenedione [45]. (The green and blue rectangles indicate the pyrene and 9-methylanthracene signals, respectively). The scale is normalized to the base peak.

## Data Availability

Please, find the Appendix A. For the moment, no additional datasets are publicly available online.

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
