# Peer review of "Evaluation of an Ozone Chamber as a Routine Method to Decontaminate Firefighters’ PPE"

_ijerph, 2021, doi:10.3390/ijerph182010587_

Round 1
Reviewer 1 Report
The harmful impact of chemicals generated during a fire on the health of firefighters is a crucial and very important problem. The results of the research published in scientific journals present the concentration of compounds in the cabins of firefighting vehicles and technical and social rooms of rescue and firefighting units. The type of substances emitted, and their concentration depend on the type of material burned and the fire conditions. In the manuscript presented for review, the Authors did not explain why they chose these two compounds from the PAHs group for research and on what basis they selected the concentration of contaminants in the experiments. It is worth extending the bibliography to include publications in which these issues are discussed, e.g., Stec, A.; Hull, T. Fire Toxicity; Oxford CRC Press: Cambridge, UK, 2010, Navarro, K. et al.; Wildland firefighter smoke exposure and risk of lung cancer and cardiovascular disease mortality. Environ. Res. 2017, 173, 462–468, Bralewska, K.; Rakowska, J.; Concentrations of Particulate Matter and PM-Bound Polycyclic Aromatic Hydrocarbons Released during Combustion of Various Types of Materials and Possible Toxicological Potential of the Emissions: The Results of Preliminary Studies. Int. J. Environ. Res. Public Health 2020, 17, 3202.
Please explain in the article content:
- Why were pyren and 9-methylanthracene selected for analysis? How does this relate to their carcinogenic, mutagenic and toxic potential?
- What was the guideline for selecting the concentrations of pyrene and 9-methyl anthracene solutions?
Technical Comments:
- Linguistic and grammar errors should be corrected.
- Punctuation marks are placed not before, but after brackets with reference.
Example:
Is: Besides being responsible for the largest share of firefighters’ deaths at work,[1] it exposes them to a large variety of chemicals,[2] including toxic fire residues that can cause permanent long-term damage and are associated with some types of cancer.[3]
Should be: Besides being responsible for the largest share of firefighters’ deaths at work [1], it exposes them to a large variety of chemicals [2], including toxic fire residues that can cause permanent long-term damage and are associated with some types of cancer [3].
Author Response
We thank reviewer 1 for her/his revision and constructive comments.
Please find our responses in the attached document.

Reviewer 2 Report
The paper by Marcella A. de Melo Lucena et al. regards the evaluation of the efficiency of the ozonolysis as a method to decontaminate PAH on firefighters’ personal protective clothes.
Authors presented and critically discussed a lot of results from different methods, that are interesting and also well compared, so the manuscript can deserve publication on International Journal of Environmental Research and Public Health. Nevertheless, the following minor revisions are required:
1) Even if interesting and generally appropriate, the Introduction section is too long and should be reduced in those parts that are too generic.
2) Figure 1: What is reported in the y-axis should be “% Transmittance”, not “Reflectance”. Please correct.
3) The same comment applies to the FTIR spectra in the supplementary.
4) Could the authors furnish, at the end of the Conclusions section, suggestions on how to revise and further investigate the use of ozone chambers as preventive measure to clean and decontaminate the firefighter’s PPE?
Author Response
We thank reviewer 2 for her/his revision and constructive comments.
Please find our responses in the attached document.
